# Scrutinizing the Impact of Alternating Electromagnetic Fields on Molecular Features of the Model Plant *Arabidopsis thaliana*

**DOI:** 10.3390/ijerph19095144

**Published:** 2022-04-23

**Authors:** Sonja Michèle Schmidtpott, Saliba Danho, Vijay Kumar, Thorsten Seidel, Wolfgang Schöllhorn, Karl-Josef Dietz

**Affiliations:** 1Department of Biochemistry and Physiology of Plants, Bielefeld University, 33615 Bielefeld, Germany; sonja.michele_schmidtpott@uni-bielefeld.de (S.M.S.); vijusaini87@hotmail.com (V.K.); thorsten.seidel@uni-bielefeld.de (T.S.); 2Department of Training and Movement Science, Mainz University, 55122 Mainz, Germany; saliba.danho@gmx.de (S.D.); wolfgang.schoellhorn@uni-mainz.de (W.S.)

**Keywords:** abiotic stress acclimation, *Arabidopsis thaliana*, electromagnetic fields, metabolites, photosynthesis, transcriptome

## Abstract

Natural and anthropogenic electromagnetic fields (EMFs) are ubiquitous in the environment and interfere with all biological organisms including plants. Particularly the quality and quantity of alternating EMFs from anthropogenic sources are increasing due to the implementation of novel technologies. There is a significant interest in exploring the impact of EMFs (similar to those emitted from battery chargers of electric cars) on plants. The model plant *Arabidopsis thaliana* was exposed to a composite alternating EMF program for 48 h and scrutinized for molecular alterations using photosynthetic performance, metabolite profiling, and RNA sequencing followed by qRT-PCR validation. Clear differences in the photosynthetic parameters between the treated and control plants indicated either lower nonphotochemical quenching or higher reduction of the plastoquinone pool or both. Transcriptome analysis by RNA sequencing revealed alterations in transcript amounts upon EMF exposure; however, the gene ontology groups of, e.g., chloroplast stroma, thylakoids, and envelope were underrepresented. Quantitative real-time PCR validated deregulation of some selected transcripts. More profound were the readjustments in metabolite pool sizes with variations in photosynthetic and central energy metabolism. These findings together with the invariable phenotype indicate efficient adjustment of the physiological state of the EMF-treated plants, suggesting testing for more challenging growth conditions in future experiments.

## 1. Introduction

Magnetic fields (MFs) and electric fields (EFs) are ubiquitous in the environment. They are of natural and anthropogenic origin. Among the natural sources is the Earth’s natural MF, known as the geomagnetic field (GMF). It includes the so-called total geophysical field, composed of abiotic factors such as gravitational MF, radioactive, seismic, and geothermal fields. A static GMF with intensities of 50–60 µT is an inevitable but a complex parameter that affects plant growth and morphogenesis [1].

MFs easily penetrate the body, whereas EFs are effectively shielded by the surface of the body. Electromagnetic fields (EMFs) combine the electric field energy from geological origin in the environment and anthropogenic nonionizing magnetic fields by electric devices [2]. Low-frequency electromagnetic fields (1 Hz–100 kHz) may excite nerve cells [3]. However, due to the increasing occurrence of EMFs as a consequence of the expanding use of wireless technology, communication devices, transport vehicles, and technical development, to name just a few, the environment concerning MFs has changed vastly in recent decades [4,5,6,7].

The exposure to EFs and MFs and their continuous intensity change might expose biological systems including plants to challenging situations that either positively or negatively impact their fitness and development. The interaction of plants with electromagnetic radiation depends on parameters such as the frequency and amplitude of the EMF [8], as well as the type, shape, and density of the organism. The immobility of the plants provides a rather fixed orientation within an EMF. Further on, the morphology of plants is characterized by a high proportion of cells that directly interact with the environment and a high surface area to volume ratio [9]. 

Applying a MF before germination of seeds promotes subsequent plant growth in some studies [10,11,12], while other approaches show growth inhibition [13,14]. It has also been reported that metabolic changes in response to EMFs occur in seeds during germination [15]. Further research demonstrated that a MF reduces oxidative damage by free radicals in plants by inducing defense enzymes. The treatment influences the activity of antioxidant enzymes, such as peroxidases and superoxide dismutase (SOD), and increases the activity of free radical ions [16,17]. Organisms and cells contain endogenous magnets, i.e., electrical charges involved in biochemical processes. With the help of the resonance behavior, a MF can interact with these electric fields [18,19]. One result is an interaction with ion currents altering the cellular ion homeostasis and the osmotic relations between both sides of the cell membrane by increasing the membrane permeability [20].

Furthermore, effects on photosynthetic light processes, and thereby on the efficiency of the photosystem II (PSII), have been reported. It was shown that low MF frequencies from 7 Hz–20 Hz decrease nonphotochemical quenching and influence the quantum yield of PSII in wheat [21]. Plants respond to different MFs/EMFs by changing their gene expression and even by varying their phenotype [22]. In that study, tomato seeds were exposed to a static magnetic field strength of 100 mT for half an hour representing a particular setting. 

Overall, the mode of action of EMF signals on plants remains poorly explored, and further experiments with different settings are urgently required. Therefore, it is crucial to further investigate interactions between EMFs and plants and their effect on plant physiological processes [13,14] in order to understand their response, possibly increase plant productivity, and enhance their resistance to such stressful situations, for example by increasing photosynthetic performance, water and nutrient uptake, and oxidative stress defense.

This research project was driven by the need to understand EMFs as a bivalent stressor on plant physiology. The benefit of using EMFs on plant organisms has been discussed for many years, and they might influence biological processes [23]. This work focuses on the impact of low-strength EMFs, generated by a low-frequency (10 and 160 Hz), low-voltage (2.5 to 10 peak to peak voltage; Vpp) alternating current (AC) after 48 h exposure on *Arabidopsis thaliana* (*A.t.*) plants in the vegetative phase of the rosette stage. The choice of EMF strength (11.5–115 µT) used was influenced by previously studied positive and/or negative effects of low-strength EMF in different biological systems, lack of comprehensive biomolecular characterization in plants, and present day’s ubiquitous presence of diverse electronic and electric devices generating similar ranges of EMFs [17,24,25]. Therefore, the study aimed to monitor (1) phenotypic changes, (2) alterations in photosynthetic parameters such as ferredoxin (Fd), photosystem I (P700), and plastocyanin (PC); (3) the transcriptome, and (4) metabolome. Using this approach, this work attempted a broader omics analysis for a detailed assessment of EMF effects on plants.

## 2. Materials and Methods

### 2.1. Plant Material and Growth Conditions

*A. thaliana* seeds of the ecotype Columbia (Col-0) were germinated on soil (special substrate, SM Max Planck Cologne, Stender^®^ AG, Papenburg, Germany) supplemented with long-term start fertilizer (Osmocote start; Nordhorn, Germany) and stratified for three days at 2–4 °C in the dark. After transfer to a growth chamber, plants grew for four weeks. The growth conditions were set to a day/night regime of 10/14 h with a temperature of 21/19 °C, a relative humidity of 55%, and a light intensity of 100 µmol photons m^−2^ s^−1^. Five plants grew in each pot, and the watering was done regularly with deionized water.

### 2.2. Exposure to Electromagnetic Field Conditions

After 4 weeks of growth, uniform *A. thaliana* plants were exposed to an EMF generated using an alternating current possessing regular repetition of multiple AC waveforms with varying peak-to-peak voltage (Vpp; 2.5 to 10) (Appendix A) for 48 h under the same growth chamber conditions (Figure 1). The used AC waveforms are a common feature of diverse electric/electronic devices carrying digital signals [26]. With the experimental setting of maximally 0.6 A and 3.1 W and the used instrumentation, the mean magnetic flux density was in the range of 45 µT, but it varied during the program between 11.5 and 115 µT.

The frequency generator PCGU1000 (2 MHz) was purchased from Velleman Group (Gavere, Belgium), and the Tesla coil plate was purchased from Schwille Elektronik (Munich, Germany). The treated plants were placed on the Tesla coil plate, and the control plants were shielded from the electric fields in the same growth chamber. The schematic experimental setup is shown in Figure 1.

### 2.3. Visualization of Phenotype

To visualize the *A. thaliana* phenotypes after EMF treatment, top view images were taken using a standard digital camera (Canon PowerShot G1X MarkII). 

### 2.4. Determination of Photosynthetic Parameters

Dark-adapted plants were exposed to 160 µmol photons m^−2^ s^−1^ for 17 s, and the redox changes of plastocyanin (PC), photosystem I (PSI; P700), and ferredoxin (Fd) were measured in intact leaves along with chlorophyll a fluorescence using a Dual-KLAS-NIR spectrophotometer (Walz, Germany) [27]. These measurements were carried out 48 h after exposure to the EMF, where the dark adaptation period was overnight and included in the treatment time. PC, P700, and Fd redox states were assessed by determining selective absorption changes of these components in the form of differential model plots [27]. The details pertaining to NIR–KLAS methodology and calculations of the derived parameters can be found in [28]. The experiment was performed with four independent biological and nine technical replicates (*n* = 36). The PSII activity-related parameters were derived from chlorophyll a fluorescence as per following equations:(1)Maximum PSII efficiency in light:(FV′/Fma)=(Fma−Fo′)Fma
(2)Operating PSII efficiency in light:(Fq′/Fma)=(Fma−Ft)Fma
(3)Coefficient of photochemical fluorescence quenching:qP=(Fq′/Fv′)(Fo′/Ft)
(4)Estimation of open PSII reaction centers:qL=(Fq′/Fv′)(Fo′/Ft)
where *F_o′_* represents the minimum fluorescence reached on switching off the actinic light, $F_{m_{a}}$ is the maximum fluorescence recorded on the onset of actinic light, and *F_t_* is the state of fluorescence at the end of the light period.

### 2.5. Isolation of Total RNA and Sodium Acetate Precipitation for Gene Expression Analysis

Leaves of five plants were harvested after 48 h exposure to electric fields (2.2) and transferred directly into liquid nitrogen. The frozen leaves were ground in presence of liquid nitrogen. A total of 100 mg of the resulting plant powder was used for isolation of RNA with the help of the RNeasy Plant Mini Kit followed by an on-column DNase digestion using RNase-free DNase set (Qiagen N.V., Düsseldorf (Hilden), Germany), in accordance with the manufacturer’s protocols. RNA was eluted with 25 µL RNase-free water and centrifuged at full speed for 1 min followed by sodium acetate precipitation. The RNA sample was filled up to 250 µL with RNase-free water, and 50 µL 3 M sodium acetate (pH 5.2) was added. After gentle inversion, incubation at −80 °C for 1 h, centrifugation at full speed and 4 °C for 20 min, the pellet was carefully washed with 70% ethanol and dried completely and dissolved in 15–20 µL RNAse-free water. The solubilization was supported by incubation at 52 °C for 10 min on a thermo-vortex-plate. To verify RNA yield and purity, RNA was measured using a nanophotometer (Implen, Westlake Village, CA, USA) and stored at −80 °C.

#### 2.5.1. RNA Sequencing 

RNA sequencing was performed by CeGaT GmbH (Tübingen, Germany). Independent biological triplicates of each treated (treatment: T) and untreated (control: C) sample of isolated RNA (2.5) were sequenced on a NovaSeq6000. Samples were quality controlled, followed by generation and quality control of a sequencing library. The read length was 2 × 100 bp, and the output was 50 M clusters (10 Gb) per sample. Illumina bcl2fastq (2.20) was used to demultiplex the sequenced reads. Adapter trimming was performed with Skewer (version 0.2.2) [29]. The trimmed tube reads were blasted against the A. thaliana genome (derived from TAIR10, version 10 released in 2010) and aligned. The evaluation of the differential expression among the sample groups was done with the DESeq2 package (version 1.24.0) [30] in R (version 3.6.1) [31]. DESeq2 uses a generalized linear model to evaluate the counts per gene and a negative binomial distribution for the error term modelling.

From the mapping, the number of mapped reads was obtained for each GeneID. The normalized values were calculated from these numbers. First, DESeq2 calculated a fictitious “reference sample”, which is defined as the geometric mean for each gene over all samples, regardless of their group membership. The counts for each gene and sample were determined and divided by the reference value. Next, the median over all genes was calculated from these ratios for each sample (size factor). For each gene and sample, the number of reads were divided by the size factor. The normalization compensated for the different library sizes and distortion effects. To illustrate the relationship among samples, hierarchical clustering and principal component analysis (PCA) were performed. The rlog-transformed data were used to gain a stable variance so that all genes were evenly distributed between the contributing samples, and the Euclidean distance of the rlog transformed data was used to calculate the distances between the samples. The log2-fold change (FC) was calculated from the normalized read numbers, and a Wald test was performed to obtain the *p*-value as the statistical indicator for the significance of the log2-fold change. For correction of multiple tests, a Benjamini–Hochberg correction was used by DESeq2 with default settings [32]. The resulting p_adj_ value was used to determine significantly differently expressed genes. The information on FASTQ files was obtained with the FastQC program (version 0.11.5-cegat) [33]. Illustrations were created with the packages ggplot2 [34] and extended [35] in R (version 3.6.1) [31].

#### 2.5.2. Transcriptome-Based Gene Identification and Quantification by qRT-PCR

RNA (2 µg) in a total volume of 11 µL was mixed with 2 µL of oligo dT primers and incubated for 10 min at 70 °C. Subsequently, 5 µL of 5 × MMLV buffer (Promega, Mannheim, Germany), 5 µL of 10 mM dNTPs, 0.75 µL of RNasin RNase inhibitor (40 U/µL, Promega, Mannheim, Germany), 1 µL of MMLV reverse transcriptase (Promega, Mannheim, Germany), and 2.25 µL of ddH_2_O were added, and the samples were incubated for 1 h at 42 °C, with subsequent stopping of the reaction at 70 °C for 15 min.

For quantitative RT-PCR, 2 µL of a 1:13 diluted cDNA was added as template to a 96-well plate (SP-0074, BioRad, Munich, Germany). The mixture was supplemented with 10 µL of KAPA SYBR FASTqPCR Mastermix (BioRad, Munich, Germany), 0.4 µL of forward and reverse primers, and 7.2 µL of ddH_2_O. The plate was sealed with a qPCR Seal Film (PeQlab, Erlangen, Germany) and briefly centrifuged in a Perfect-SpinP centrifuge (PeQlab, Erlangen, Germany). Subsequently, qRT-PCR was started in the MyiQTM Optics Module Thermocycler (BioRad, Munich, Germany). The primer sequences (Appendix A) were designed using Primer 3.0 software [36]. PCR conditions were: 2 min at 95 °C, 35 cycles of 15 s at 95 °C, 1 min at 60 °C, and 1 min at 95 °C, and all runs were followed by a melting curve analysis with a temperature gradient between 60 and 90 °C in 1 °C steps. Previously, the amplification efficiencies, primer specificities, and linear range for each primer pair were tested using seven dilution levels of pooled cDNA from all samples of the experiment. 

The two different reference genes *ACT2* (AT3G18780, *Actin*) and *UBI10* (AT4G05320, *Ubiquitin 10*) served as normalization reference and used as per the algorithm of Pfaffl [37].

Relative expression was calculated from the raw fluorescence data using LinRegPCR 11.0 software. Individual PCR efficiencies and threshold cycle (Ct-values) were determined after each cycle and used to determine relative expression according to Vandesompele et al. [38]. Geometric mean values of ACT2- and UBI10-mRNA levels served as normalization factors, and the comparisons were based on the expressions of the untreated samples. Measurements were performed in duplicate of samples obtained in four independent experiments for each treated and untreated sample.

#### 2.5.3. Gene Onotology (GO) Analysis 

Deregulated transcripts in EMF-treated plants were subjected to GO term analysis using the online Panther tool [39,40]. To this end, the number of transcripts with increased abundance in EMF-treated plants exceeding 2-fold or decreased abundance below 0.50-fold were compared to the Arabidopsis reference database for relative determination of over or underrepresentation for different GO terms [39,40]. 

### 2.6. Metabolite Profiling

Metabolome analysis was performed at CeBiTec (Center for Biotechnology, Bielefeld University, Bielefeld, Germany). The plant material was pulverized in liquid nitrogen, and 80–100 mg was transferred into 2 mL reaction tubes. The plant powder was freeze-dried for 24 h in an RVC 2–18 freeze-drying unit (Martin Christ Gefriertrocknungsanlagen GmbH, Osterode am Harz, Germany). A total of 5 mg dry weight was transferred to tubes, and 10–20 silica beads (1 mm, Roth, Karlsruhe, Germany) were added. Extraction was performed by adding 1 mL of 10 mM ribitol in 80% (*v*/*v*) methanol followed by homogenization using a Precellys 24 cell mill (Bertin Technologies, Frankfurt am Main, Germany) for three times 1 min at 6200 rpm with 5 s pause each time. Samples were centrifuged at 14,680 rpm for 20 min, and the supernatant was transferred to reaction vessels. Under nitrogen gassing, samples were evaporated in the derivatization block (Reacti-Therm Dry Block Sample Incubation System, Thermo Scientific Pierce Protein Biology, Waltham, MA, USA). After 45–60 min, derivatization was performed by adding 75 µL of derivatization solution (20 mg/mL methoxylamine hydrochloride in pyridine) followed by incubation for 1.5 h with stirring at 37 °C. Subsequently, 75 µL of MSTAFA (N-methyl-N-(trimethylsilyl) trifluoroacetamide) was added, and the sample was incubated for another 30 min. After centrifugation at 4000× *g* at room temperature for 5 min, the supernatant was transferred to a 150 µL micro insert of a 1.6 mL threaded vial and sealed.

The measurements were performed using an ion trap GC–MS (gas chromatography–mass spectrometry) system of the Trace GC Ultra gas chromatography and the ITQ 900 ion trap mass spectrometer (Thermo Scientific, Waltham, MA, USA).

MS analysis was performed using Xcalibur 2.0.7 software (Thermo Scientific, Waltham, MA, USA). The metabolites were identified using the retention index [41] and the additional adaptation of mass spectra to the previously measured reference substances [42]. The quantifications were normalized to ribitol and the sample dry weight.

Measurements were performed in duplicate in six independent biological experiments for each treated and untreated sample.

### 2.7. Statistical Analysis

The data represent the means ± standard error (SE) of at least three independent experiments as indicated. Each sample consisted of pooled tissues from 5 plants per pot. Student’s *t*-test analysis was conducted to analyze the statistical significance of difference between the EMF-treated and control plants. In the case of more than two samples, a one-way or two-way analysis of variance (ANOVA) was performed, respectively, and a Student‘s *t*-test was applied to detect the statistical significance of the difference between the individual data sets. Asterisks indicate statistically significant differences (*p* < 0.05/*p* < 0.01).

## 3. Results

*Arabidopsis thaliana* was selected for the experiments due to its unique position as the best explored model plant. All plants were grown in the same growth chamber under constant conditions adjusted to a fixed day/night cycle, temperature, and humidity thereby providing a homogenous starting material. At the age of about 30 days, plants were in the rosette stage and exposed to EMFs (Figure 1). The control plants were placed outside the irradiation zone in the same growth chamber guaranteeing identical environmental conditions for the control and treated plants apart from the EMF. In light of the identical starting material and the same incubation conditions, we expected a high sensitivity for stimulus-induced changes.

### 3.1. The Variation of the Phenotype

After 48 h of irradiation, the control and treated plants lacked significant phenotypic differences (Appendix A). Usually only strong acute stress effects such as heat shock, pathogen infection, or freezing express strong phenotypic changes within 2 days of treatment [43,44,45]. The dry weights were 1.13 ± 0.08 g/rosette in the control plants and 1.10 ± 0.14 g/rosette in the treated plants (*n* = 5, with 5 rosettes each). To elaborate on long-term effects, a 3-week irradiation was carried out, with the plants being transferred to irradiation at an early developmental stage, and their phenotype was documented on every seventh day. The final assessment occurred after 21 days and revealed no significant differences between the control plants and the treated plants (Appendix A). Thus, no reliable phenotypic changes could be observed in both test batches (Appendix A).

### 3.2. The Response of Photosynthetic Parameters to Electromagnetic Fields

To scrutinize the influence of EMFs on the efficiency of the photosynthetic electron transport chain (PET), kinetic redox changes of the PET components plastocyanin (PC), photosystem I (P700), and ferredoxin (Fd) were recorded in intact leaves from 30-day-old plants. In parallel, chlorophyll a fluorescence-derived PSII activity was monitored, and the quantum yield was calculated. The plants were treated for 48 h with EMFs (treatment, T) or kept under control conditions (control, C). They were dark-adapted before being exposed to 160 µmol photons m^−2^ s^−1^ for 17 s. During light exposure PET activity was measured and compared to the untreated control. 

The kinetics of oxidative changes in percent and the chlorophyll a fluorescence as relative units were detectable upon administration of actinic light. The 17 s time span was flanked by dark periods of 3 s and ~300 ms duration before and after illumination, respectively (Figure 2A,B). The data revealed that the components on the acceptor side of PSI (Figure 2A), namely PSII activity and redox kinetics of PC, showed stronger alterations upon EMF exposure than those of PSI itself and Fd acting downstream of PSI.

A closer look at the redox kinetics data dissected the underlying processes further. Multiple phases occurred after the onset of light. The initial oxidation of PC and PSI and reduction of Fd proceeded fast both in the EMF-treated and control plants. The subsequent slow changes of the redox states revealed differences, while the changes upon the light-to-dark shift at the very end were kinetically not resolved. In this context, it was evident from Figure 2 that for all components (PSII, PC, PSI, and Fd), the strongest divergence in behavior appeared in the late phase of illumination before the light was switched off. 

The progressively reduced signal deflection with light exposure time appeared due to activation of photochemical reactions downstream of the light absorption at the photosystem reaction centers. The clear differences in the fluorescence kinetics between the treated and control plants (Figure 2A) indicate either less nonphotochemical quenching or a higher reduction of the plastoquinone pool or both. The derived PSII-related parameters, i.e., maximum (Fv’/Fm’) and operating (Fq’/Fm’) PSII efficiency in light, the coefficient of photochemical fluorescence quenching (qP), and the estimated fraction of open PSII reaction centers (qL) all indicate a significantly lowered value, i.e., 10, 32, 24, and 28%, respectively, due to EMF irradiation (Figure 3A). 

The Dual-KLAS-NIR permitted the automated estimation of PC, PSI, and Fd pool sizes and therefore deconvolution and quantification of their individual redox change. The efficiency of photosystem I depends majorly on the balance between the pool sizes and the rate of redox changes of the acceptor (PC) and donor side (Fd) components for the P700 reaction center. The treated plants showed a significant (*p* < 0.05) decrease in PSI pool size and consequently increases in the PC/PSI and Fd/PSI ratios (Figure 3C). A comparison of redox states for PC, PSI, and Fd at apparent steady state before the light was switched off showed significantly higher oxidation (1.86-fold: PC, 1.53-fold: PSI) for the control plants in comparison to those exposed to EMFs, while no major differences were observed for Fd (Figure 3D). 

### 3.3. Effect of Electromagnetic Fields on the Leaf Transcriptome

After the physiological analysis of photosynthesis, we aimed at scrutinizing other molecular alterations in EMF-exposed plants by transcriptome analysis. RNA sequencing was conducted with samples from three independent experiments. The quality control of all RNAs showed yields between 3.5 µg and 4.5 µg with mean concentrations of 170 ng/µL and an RNA integrity number (RIN) of about 7. After sequencing, 295,041,000 raw paired reads were obtained with an average of 49.17 million reads and 10 Gb base sequence information per sample (Appendix A). Read lengths ranged between 100 and 102 nt (Appendix A). The sequence qualities of the trimmed FASTQ reads offered the highest average base quality per read at a Phred score of 36 (Appendix A), and the highest GC content per read between 39% and 44% (Appendix A). 

An overview of changes in gene expression is illustrated as hierarchical clustering and a PCA plot (Appendix A). The hierarchical clustering of the samples was performed according to the similarity of their expression data based on all genes that had received at least one read in at least one sample, and the samples were stained by group (control/treatment). The representatives from the two groups did not cluster together (Appendix A). The graphics of the principal component analysis showed the two dimensions of the data space, which contributed most to the intersample variance. The percentages on the axes describe the proportion of this central component of the total variance. A group formation is not recognizable (Appendix AB). 

The trimmed raw reads were aligned against the *A. thaliana* genome. From the mapping, we obtained the number of mapped reads for each geneID. The analysis of differentially expressed genes (DEGs) is listed in Appendix A and shows 23,965 transcripts (Appendix A). RPKM are the reads per kilobase of transcript per million mapped reads of C1-C3 (control) and T1-T3 (treatment). The p_adj_ displays the false discovery rate (FDR) corrected *p*-value, and for all transcripts, the value is 0.999 (Appendix A), indicating the absence of significant DEGs. 

Altogether, the clustering, PCA-plots, and p_adj_-values showed that the RNA sequencing analysis failed to reliably identify many DEGs between the control and treatment groups based on the adjusted *p*-value, possibly indicating that the variation between the independent growth regimes and treatments exceeded the effects of the EMF exposure. The gene ontology analysis of transcripts with ≥2-fold increased or ≤0.50-fold decreased abundance identified two overrepresented groups, namely microtubule-associated transcripts with 2.44-fold enrichment and the group of unclassified transcripts with 1.42-fold enrichment (Table 1). All other GO terms were either not significantly affected or depicted decreased representation relative to the expected number. Among these underrepresented GO terms were several chloroplast and photosynthesis-related transcripts. For this reason, we went for a general quantitative real-time PCR (qRT-PCR) validation of certain deregulated transcripts with a higher magnitude of change as a proof of concept and did not focus on photosynthesis.

### 3.4. Real-Time Quantitative PCR

A qRT-PCR was used to quantify individual transcripts that showed an upregulation due to the values-based mean higher than > 10-fold, *p*-value < 0.05, and log2FC > 1. We hypothesized that transcripts with a major change, even if statistically insignificant according to p_adj_, may still be valid and possibly can be validated by deeper analysis. Therefore, based on the global transcriptome analysis results, settings were chosen to pinpoint the most effective upregulated transcripts. Sixteen chosen transcripts showed an upregulation in response to the EMF treatment (Appendix A).

Eight transcripts were selected out of these, which were among the strongest differentially regulated transcripts in the transcriptome analysis of the control (C) and treatment (T) groups (Appendix A). To validate the qRT-PCR primer systems (Appendix A), a qPCR reaction with a cDNA concentration series was carried out. The used primers showed a slope between −2.6 and −3.3 with Pearson correlation coefficients (R^2^) between 0.949 and 0.998. Only HSD5 showed a less optimal slope of −1.7 and R^2^ = 0.872 (Appendix A). To check the PCR products, melting curves were recorded at the end of each qPCR in the range between 60 °C and 90 °C. The melting curves showed a maximum of the negative, first derivative of the relative fluorescence (−Δ(RFU)_dT) of 600 to 900 in a temperature range between 80 and 85 °C. Only HSD5 showed two peaks at 400 −Δ(RFU)_dT (Appendix A).

Figure 4 shows the results of the qRT-PCR with relative expression levels depicted. The qRT-PCR was used to check the differential expression patterns of the following eight transcripts: BOR5 (AT1G74810, an HCO3-protein family), NTR (AT3G00720, a ‘novel transcribed region’ according to TAIR), HSD5 (AT4G10020, a hydroxysteroid dehydrogenase), RLP51 (AT4G18760, a receptor-like protein 51), SYTD (AT5G11100, a calcium-dependent lipid-binding family protein), ARD3 (AT2G26400, an acireductone dioxygenase), VIM2 (AT1G66050, a Zn-finger protein with RING and SRA domains, likely with activity as E3-ubiquitin ligase), and SNOR (AT2G20721, snoRNA). The two housekeeping genes ACT2 and UBI10 served as normalization reference.

The genes *RLP51* (1.5-fold), *SYTD* (1.75-fold), *ARD3* (0.5-fold), and *SNOR* (0.75-fold) displayed a significant (*p* < 0.01) change in the transcript level (Figure 4). Whereas *RLP51* and *SYTD* showed increased expression levels compared to the control (Figure 4), *ARD3* and *SNOR* demonstrated a decreased expression level. The transcripts *BOR5*, *NTR*, *HSD5,* and *VIM2* displayed a slight change in expression, which was not significantly different from the control (Figure 4).

### 3.5. EMF Effects on Leaf Metabolome

In comparison to the half-life times of transcripts, the turnover of most metabolites is faster and, therefore, metabolic changes may be more sensitive to functional disturbances. On the other hand, the regulation of gene expression may also occur within seconds, while metabolite pools may be subjected to complex balancing mechanisms counteracting the establishment of significant changes. To explore the effect of EMF on the leaf metabolome, a GC–MS-based analysis was performed. A total of 91 metabolites of different categories including sugar derivatives, amino acids, and nucleotides were reliably detected and quantified based on the external standard ribitol and normalized to dry weight (Appendix A). To ease the assessment of metabolic changes, the ratio between the treated and control extracts was calculated from the normalized metabolite contents. Of all 91 metabolites, the contents of 23 metabolites increased; 30 metabolites decreased, and 38 metabolites remained unchanged between control and treatment (ratio = 1 ± 0.1) (Appendix A). 

Among all increased metabolites, the highest metabolite content was detected for the amino acid serine. Its concentration significantly changed 1.3-fold upon treatment (Figure 5A), increasing from 151 r.u. mg^−1^ dry weight under control conditions to 190 r.u. mg^−1^ dry weight upon treatment (Figure 5B). In the range of 20–200 r.u. mg^−1^ dry weight, there was melibiose with a significant change of 1.7-fold, and there were glucose, glutamate, and L-aspartate with contents between 20 and 100 r.u. mg^−1^ dry weight, but no significant change between treatment and control (Figure 5A,B). Lower metabolite contents (2–10 r.u. mg^−1^ dry weight) were detected for arabinose, fructose, glyceraldehyde, lactate, leucine, myo-inositol-phosphate, and xylose, which also revealed significant increases upon EMF treatment, except for leucine and xylose (Figure 5C). The fructose amount was 10 r.u. mg^−1^ dry weight in the treated sample and 6 r.u. mg^−1^ dry weight in the control corresponding to a 1.6-fold increase (Figure 5A,C). An increased abundance was also recorded for glyceraldehyde (3 vs. 5 r.u. mg^−1^ dry weight of control/treatment, fold change of 1.8, Figure 5A,C) and myo-inositol-P-derivatives (2 vs. 1.6 r.u. mg^−1^ dry weight of control and treatment value, fold change of 1.3, Figure 5A,C). 2-isopropylmalate, dihydroxyacetone phosphate (DHAP), galactose, and maleic acid showed the lowest content between 0 and 1.5 r.u. mg^−1^ dry weight and significant fold change. Further, cis-aconitate, erythrose-4-phosphate, gluconate-1.5-lactone, lysine, phosphoenolpyruvate (PEP), tyrosine, and urea showed low contents, but no significant fold change (Figure 5D).

Metabolites with decreased abundance in treated leaves are depicted in Figure 6A. Of the 30 downregulated metabolites, synaptic acid showed the least change with a fold change of 0.88, and homoserine showed the highest decrease of 0.4-fold. The remaining 28 metabolites ranged between these two ratios (Figure 6A). The highest relative metabolite content was in a range of 15–100 r.u. mg^−1^ dry weight and 400 r.u. mg^−1^ dry weight (Figure 6B), and a significant fold change (*p* < 0.01) was seen for glutamine with 23 r.u. mg^−1^ dry weight (control) and 11 mg^−1^ dry weight (treatment); glycerate (18 vs. 12 r.u. mg^−1^ dry weight); glycine with a value of 100 r.u. mg^−1^ dry weight (control) and 55 mg^−1^ dry weight (treatment); and sucrose with the highest control value of 445 r.u. mg^−1^ dry weight and a treatment value of 335 r.u. mg^−1^ dry weight. In this group of the highest relative metabolite content was also sinapic acid (18 vs. 16 r.u. mg^−1^ dry weight) but with no significant change in the ratio of treatment and control (Figure 6B). The next ten metabolites showed mostly significantly lower metabolite contents between 1 and 10 r.u. mg^−1^ dry weight (Figure 6C); e.g., glycolate, homoserine, phenylalanine, shikimate, pyruvate, and succinate were present at low metabolite contents, and the data revealed significant downregulation. The last group contained metabolites with low content in leaves (0–1 r.u. mg^−1^ dry weight) and included 15 metabolites (Figure 6D) of different pathways such as amino acids, sugars, and carboxylic acids, e.g., asparagine, homocysteine, maltose, fructose-6-phosphate, glucose-6-phosphate, glycerate-2-phosphate, sedoheptulose-7-phosphate, glucuronic acid, and α-ketoglutarate. Except for 2-methylcitrate, fructose-6-phosphate, glucose-6-phosphate, glycerate-2-phosphate, and α-ketocaproate all other metabolites showed significant differences in the downregulation of the metabolites between control and treatment (Figure 6D). Overall, EMF exposure caused several significant changes in the metabolome.

## 4. Discussion

The applied EMF program aimed to mimic a mixed setup of natural and anthropogenic components. In contrast to the Earth’s static magnetic field, the alternating EMF administered at two different frequencies and three application patterns (rectangular, sine, and triangular shape) constantly changed the EMF environment of the plants. The phenotype of the plants, such as size, dry weight, and coloration, remained unchanged during the 48 h exposure used for all analyses and also during a 3-week exposure. However, each physiological and molecular parameter scrutinized in the treated and control plants revealed sensitivity to the EMF and the initiation of acclimation responses. 

### 4.1. Changes in Photosynthetic Parameters 

Photosynthetic primary reactions are known to display magnetic field dependencies [46]. Klevanik observed changed fluorescence emission from trimeric PSI particles isolated from *Synechococcus elongatus* exposed to a changing magnetic field with a sweep rate of 825 Gauss/s and emitted from a 600 W ramp generator. EMF exposure affects the mobility of the primary radical pair P700^+^ A_0_^−^. The analytical MF applied in such studies differs in frequency and exceeds the strength of EMF treatment in our study by orders of magnitude. This is valid for all paramagnetic resonance (EPR) and Electron-Nuclear-Double-Resonance (ENDOR) spectroscopy studies [47] where GHz frequency is applied. Nevertheless, these studies reveal the presence of EMF-sensitive structures in photosynthetic organisms. 

Here we used extremely low-frequency EMFs of 10 and 160 Hz, sometimes abbreviated ELFMFs. ELFMFs generate resonance at the Schumann resonance frequencies and lower nonphotochemical quenching parameters such as NPQ_max_ or NPQ_F_ in wheat and pea during illumination even after a short application of 30 min at 18 µT [21]. The effects were species-dependent. These results are in line with the lowered photosynthetic efficiencies detected for *A. thaliana* in this study (Figure 3).

Our analysis also considered the effects on pool sizes and redox states of components of the photosynthetic apparatus, such as ferredoxin (Fd), plastocyanin (PC), and PSI (Figure 2 and Figure 3). In particular, the molar ratios between PC/PSI and Fd/PSI had changed significantly after the 48 h exposure to the EMF. Apparently, the molecular expression, translation, assembly, and degradation systems that control the PET rate respond to the EMF and adjust individual components. It should be noted that the PET and chloroplast metabolism are considered as environmental sensory systems regulating nuclear gene expression and assembly processes [48,49]. Searching for published studies on EMF effects on single PET components was unsuccessful. Importantly, the adjustment realized a similar redox state downstream of photosystem I. However, as discussed below, effects were seen on metabolites including those with a role in the Calvin–Benson cycle. Thus, EMF interferes with the state of the photosynthetic light and dark reactions.

### 4.2. Variation in Transcript Amounts May Hide a More Severe Reorganization of the Transcriptome

Transcript amounts often sensitively respond to environmental cues [50,51]. Therefore, RNA sequencing analysis was attempted with the expectation that sensitive responses of the transcriptome might provide strong indications of the reorganization of gene expression and RNA homeostasis of EMF-exposed plants. Therefore, it was surprising that the transcript variation between the independent experiments exceeded that between the treatment and the control both in the hierarchical cluster analysis and the principal component analysis (Appendix A). To elaborate on the differences at the level of the entire transcriptome, a more dependent analysis might need to be performed on identical biological materials, e.g., grown simultaneously in a random block design [52]. Nevertheless, the transcriptome data made it possible to identify selected transcripts whose amount was altered but not at a significant level according to the adjusted *p*-value. The differences between EMF-treated samples and the control plants could be confirmed by quantitative RT-PCR on four out of eight selected transcripts (Figure 4).

The alterations concerned both transcripts with increased and decreased abundance upon EMF exposure, proving EMF effects also at the level of gene expression or RNA processing and turnover. The selected transcripts code for eight proteins with functions in diverse cellular processes such as nutrient transport (*BOR5*), specialized metabolism (*ARD3*), chemical sensing (*RLP51*), signal transduction (*SYTD*), protein turnover (*VIM2*), and RNA processing (*SNOR* and possibly *NTR*). Whether these alterations reflect nonspecific responses or changes with relevance for the acclimation to the EMF environment remains to be shown in future work. 

### 4.3. EMF-Induced Metabolic Alterations

EMF effects on metabolism should be reflected in the metabolite contents and metabolite ratios (Appendix A). Metabolite profiling documented that at this level, EMF exposure also induced significant alterations in Arabidopsis leaves. Contents of most mono-phosphorylated metabolites decreased, and this may be indicative for a decrease in the Calvin–Benson cycle and glycolytic steady state levels, albeit it must be kept in mind that steady state levels are a poor indicator for fluxes [53]. However, autocatalytic buildup and the maintenance of significant metabolite pool sizes in the Calvin–Benson cycle are a decisive feature for high rates of photosynthetic CO_2_ fixation at constant temperature and ambient CO_2_ concentration [54]. The only metabolite with increased contents upon treatment was dihydroxyacetone phosphate (DHAP) (treated: 0.0082, control: 0.0060), while 3-phosphoglycerate (G3P: treated: 0.047; control: 0.049) was unchanged. This shows that cellular energization as a product of phosphorylation potential and NADPH/NADP^+^-ratio as indicated by the DHAP/G3P-ratio was slightly increased [55]. Sedoheptulose-7-phosphate (S7P) is also of interest since S7P may be taken as a marker of the metabolites in the regeneration phase of the Calvin–Benson cycle [54]. The ratio of S7P/DHAP dropped from 7.26 to 2.98 possibly indicating limitations in the regeneration phase of the dark reactions of photosynthesis that required some readjustments to maintain proper CO_2_ fixation. The unaltered phenotype reflects the capability of the plant to efficiently deal with these modifications and to maintain overall productivity.

Such alterations also concerned sucrose as the main product of photosynthesis, which decreased by 25% upon EMF treatment. In a converse manner, amounts of both glucose and fructose increased in treated plants. It is important to note that the total sum of relative amounts of detected metabolites was almost identical with 2485 r.u. mg^−1^ dry weight in the control and 2364 r.u. mg^−1^ dry weight in the treated samples. Likewise, as described above, many amino acid contents remained unchanged, e.g., alanine with 247 r.u. mg^−1^ dry weight in the control and 246 r.u. mg^−1^ dry weight in the EMF-exposed plants. Also, the levels of carboxylic acids with a role in the citric acid cycle, but also stored in the vacuole, remained invariable except for succinate and α-ketoglutarate with a decrease of 36% and 40%, respectively. 

Another interesting change concerned the ratio of lactate (treated: 15.21; control 12.74) to pyruvate (Pyr: treated: 5.32; control: 8.63), which doubled from 1.48 to 2.86. The conversion of pyruvate to lactate is catalyzed by lactate dehydrogenase as part of fermentation and regenerates NAD^+^ for the continuation of glycolysis; however, LDH is inefficient in catalyzing the reverse reaction. Other enzymes such as glycolate oxidase 3 may oxidize lactate [56]. It appeared interesting to consider the ratio of the consecutive metabolites in glycolysis phosphoenolpyruvate (PEP: treated: 0.00038; control: 0.00022) and pyruvate (see above), resulting in a 2.8-fold higher PEP/pyruvate ratio in EMF-treated plants. Both the increased PEP/Pyr and the increased lactate/Pyr ratios might indicate an increased energization of the cytoplasm, as postulated above for the Calvin–Benson cycle. 

## 5. Conclusions

EMF exposure occurs ubiquitously on Earth and is a regular environmental parameter all life forms have to deal with. Anthropogenic alternating and rapidly changing EMF emission accentuate the requirement to research the EMF–life and, as conducted here, EMF–plant interaction. The data show that alternating EMF-exposed plants undergo significant alterations at the levels of photosynthesis, transcriptome, and metabolome. The underlying perception and response mechanisms to weak EMF still need deeper scrutiny and could involve changes in cell signaling, e.g., by interfering with Ca^2+^ fluxes and concentrations [57]. The changes in photosynthetic features are remarkable particularly since the gross growth parameters were unaltered. A profound metabolic readjustment was detectable upon EMF exposure that would merit further experiments, both in the context of photosynthesis and respiratory energy metabolism. The growth conditions with about 5% of full sunlight (100 µmol photons^.^m^−2^ s^−1^) might not be ideal to work out possible negative effects of the EMF treatment on plant performance. In future experiments, it would be worthwhile to combine such EMF treatments with challenging growth conditions, e.g., by growth in saturating light or combinatorial stress applications [58]. Future experiments could also include the study of early development such as seed germination and seedling establishment similar to [57].

## Figures and Tables

**Figure 1 ijerph-19-05144-f001:**
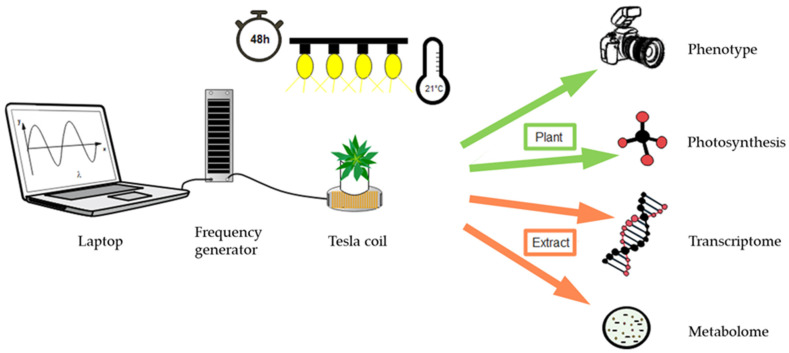
Schematic representation of the experimental setup used for electromagnetic field irradiation and analysis of plant material. The plants were grown and treated at 21 °C and 10/14 h day/night conditions. The settings for the electromagnetic irradiation were transferred to the Tesla coil via the frequency generator. The treated plants were placed on the emitter and irradiated for 48 h. The untreated plants were placed outside of the irradiated area. The intact plants were phenotypically and photosynthetically analyzed, while the transcriptome and metabolome analysis were carried out with plant extract.

**Figure 2 ijerph-19-05144-f002:**
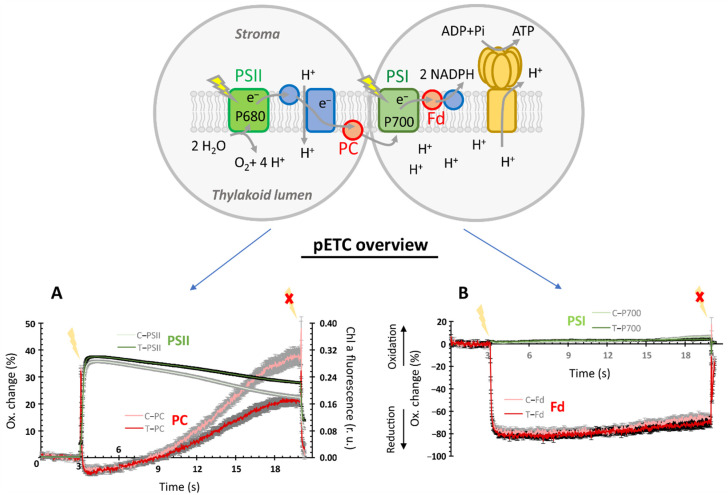
Effect of electromagnetic field irradiation on light-induced redox kinetics in *A. thaliana* photosynthetic electron transport (PET). Dark-adapted leaves of 30-day-old *A. thaliana* (Col-0) plants, which had previously been exposed to EMF irradiation for 48 h (treatment = T, control = C) were analyzed for the light-induced kinetic redox changes in PSI (photosystem I), PC (plastocyanin), and Fd (ferredoxin). Simultaneously, the chlorophyll a fluorescence was recorded to also quantify the efficiency of PSII (photosystem II). The pictogram in the figure highlights these components and their relative position in the PET. For this analysis using Dual-KLAS-NIR (Walz, Germany), leaves were exposed to light (160 µmol photons m^−2^ s^−1^) for 17 s with simultaneous recording of the electron flow through PET. This light period was flanked by recording for 3 s in the dark before and ~300 ms afterward as presented in A and B. As indicated in the figure, (**A**) gives the major treatment-specific effects on the Chl a fluorescence and redox kinetics of PC components on the acceptor side of PSI, while (**B**) shows minor changes observed for PSI and Fd redox shifts. The given redox changes for individual components are based on deconvolution using the model spectra generated from similarly grown plants followed by scaling using the estimated pool sizes for each component.

**Figure 3 ijerph-19-05144-f003:**
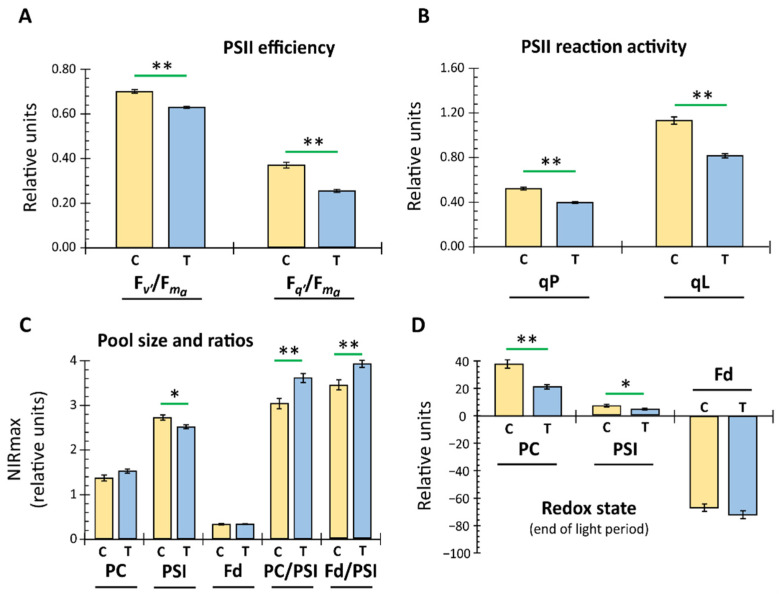
Effect of 48 h electromagnetic field irradiation on photosynthetic parameters in *A. thaliana*. The control (C) and treatment (T) groups were handled as described (2.2). The recorded data, as given in Figure 2, were further evaluated for identifying specific irradiation effects among differently derived photosynthetic parameters and their statistical significance. Parameters such as maximum PSII efficiency in light (F_v’_/F_m__a_; **A**), operating PSII efficiency in light (F_q’_/F_m__a_) (**A**), coefficient of photochemical fluorescence quenching (qP; **B**), and estimated fraction of open PSII reaction centers (qL; **B**) were calculated (means ± SE, *n* = 34–37, *p* < 0.01 **, Student’s *t*-test). Derivation of these parameters is based on the recorded initial chlorophyll a fluorescence maxima in dark-adapted leaves on exposure to actinic light (160 µmol photons m^−2^ s^−1^). Dual-KLAS-NIR (Walz, Germany) also enables automated estimation of pool sizes and their respective ratios for PC, PSI, and Fd, given as maximum absorption in the near-infrared NIRmax (**C**) (means ± SE, *n* = 34–37, *p* < 0.05 *, and < 0.01 **, Student’s *t*-test). (**D**) The recorded redox state for PC, PSI, and Fd at the end of the light period was compared between control and treated plants, and the significant differences are indicated (means ± SE, *n* = 34–37, *p* < 0.05 *, and < 0.01 **, Student’s *t*-test).

**Figure 4 ijerph-19-05144-f004:**
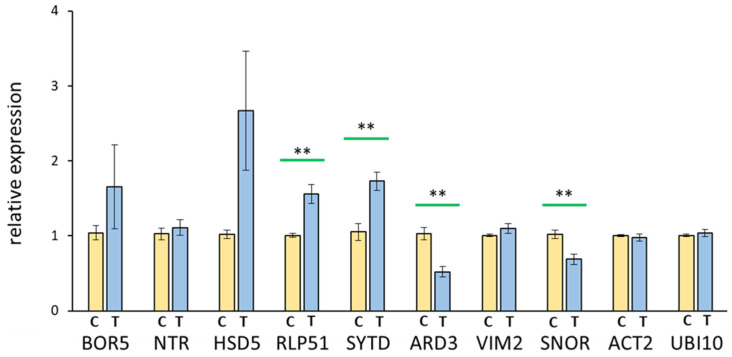
Quantitative qRT-PCR. Representation of the relative transcript amounts of the targets *BOR5*, *NTR, HSD5*, *RLP51*, *SYTD*, *ARD3*, *VIM2,* and *SNOR* (see text for details) from the treatment (T) and control (C) cDNA. The values were normalized on the control and the housekeeping transcripts *Actin* and *Ubiquitin* [37]. Relative transcript amount to control and reference genes actin and ubiquitin. Data were obtained in four independent experiments with replication, *n* = 8, means ± SE, Student’s *t*-test, and *p* < 0.01 **, indicated by green bars.

**Figure 5 ijerph-19-05144-f005:**
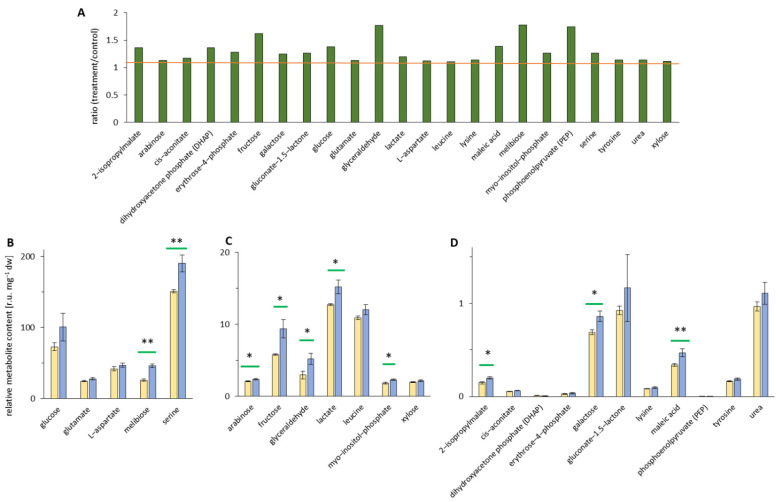
Metabolomic changes in *Arabidopsis thaliana* leaves represented as ratio of treatment/control >1.1. Metabolite levels of treatment and control extracts were measured using GC–MS in leaves from 30-day-old plants after 48 h of EMF irradiation. Control samples are given in yellow, and the treated samples are in blue. (**A**) Overview of increased ratio of the relative metabolite content between treated and control extract metabolites (>1.1, orange bar). (**B**) Upregulated high relative metabolite content [r.u. mg^−1^ dry weight]. (**C**) Upregulated intermediate relative metabolite content [r.u. mg^−1^ dry weight]. (**D**) Upregulated low relative metabolite content [r.u. mg^−1^ dry weight]. Independent experiments = 3, *n* = 6, means ± SE, Student’s *t*-test, *p* < 0.01 **, and *p* < 0.05 *, indicated by green bars.

**Figure 6 ijerph-19-05144-f006:**
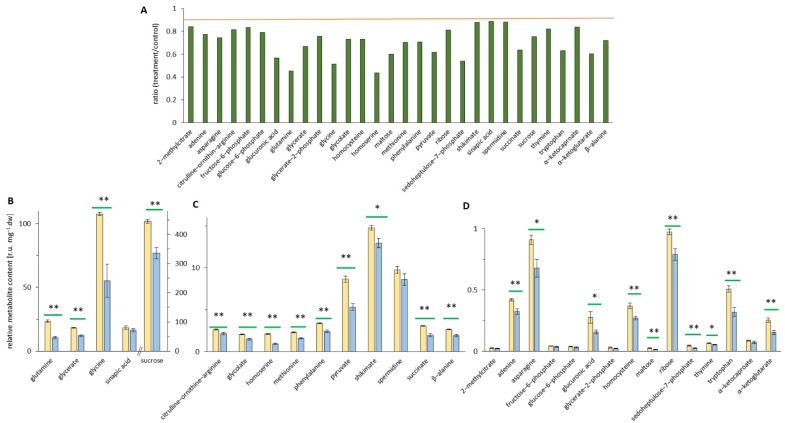
Metabolomic analysis in *Arabidopsis thaliana* plants with a ratio of treatment/control < 0.9. In 30-day-old plants (Col-0) after 48 h of EMF irradiation, the relative metabolite levels of treatment and control extracts were measured using GC–MS. Control samples are given in yellow, and the treated samples are in blue. (**A**) Overview of downregulated ratio of the relative metabolite content between treated and control extract metabolites (< 0.9, orange bar). (**B**) Metabolites present at high levels [r.u. mg^−1^ dry weight]. (**C**) Metabolites present at intermediate levels [r.u. mg^−1^ dry weight]. (**D**) Metabolites present at low levels [r.u. mg^−1^ dry weight]. Independent experiments = 3, *n* = 6, means ± SE, Student’s *t*-test, *p* < 0.01 **, and *p* < 0.05 *, indicated by green bars. There is a significant (*p* < 0.01/*p* < 0.05) downregulation of the metabolites between control (yellow) and treatment (blue).

**Table 1 ijerph-19-05144-t001:** Gene ontology (GO) analysis of EMF-induced transcript changes. (**A**) All 1114 unique transcripts with fold change higher than 2-fold were considered for the analysis. (**B**) All 975 unique transcripts with fold change values < 0.50 were included in the analysis. The table gives the GO term description, the number of transcripts belonging to the group (Ref.), the number affected by EMF, the expected and observed enrichment, and the corrected *p*-value.

A GO Cellular Component With Increased Abundance	A.t. (Ref.)	EMF Effect	Expected	EMF- Enriched	*p*-Value (FDR)
**Microtubule** (0005874)	182	18	7.39	2.44	2.78 × 10^-2^
**Unclassified** (UNCLASSIFIED)	1929	111	78.34	1.42	1.01 × 10^-2^
**Cellular component** (0005575)	25,501	1003	1035.66	0.97	1.03 × 10^-2^
**Mitochondrion** (0005739)	4385	134	178.09	0.75	9.17 × 10^-3^
**Intracellular protein-containing complex** (0140535)	710	13	28.83	0.45	3.50 × 10^-2^
**Golgi apparatus** (0005794)	1161	20	47.15	0.42	5.86 × 10^-4^
**Catalytic complex** (1902494)	1224	19	49.71	0.38	6.98 × 10^-5^
**Plastid membrane** (0042170)	476	7	19.33	0.36	4.86 × 10^-2^
**Nucleolus** (0005730)	488	6	19.82	0.3	1.59 × 10^-2^
**Endosome** (0005768)	407	5	16.53	0.3	3.74 × 10^-2^
**Membrane protein complex** (0098796)	606	7	24.61	0.28	2.49 × 10^-3^
**Thylakoid membrane** (0042651)	375	4	15.23	0.26	3.63 × 10^-2^
**Chloroplast envelope** (0009941)	601	6	24.41	0.25	7.05 × 10^-4^
**Peroxisome** (0005777)	321	3	13.04	0.23	4.05 × 10^-2^
**Chloroplast thylakoid** (0009534)	438	4	17.79	0.22	7.97 × 10^-3^
**Cytosolic ribosome** (0022626)	294	2	11.94	0.17	2.77 × 10^-2^
**Endoplasmic reticulum membrane** (0005789)	305	2	12.39	0.16	2.10 × 10^-2^
**Ribosomal subunit** (0044391)	314	2	12.75	0.16	1.53 × 10^-2^
**Chloroplast stroma** (0009570)	703	2	28.55	0.07	4.17 × 10^-8^
**B GO cellular component** **With decreased abundance**	**A.t. (Ref.)**	**EMF effect**	**Expected**	**EMF-** **enriched**	***p*-value (FDR)**
**Plasmodesma** (0009506)	880	15	31.28	0.48	4.8 × 10^-2^
**Chloroplast stroma** (0009570)	703	10	24.99	0.4	3.0 × 10^-2^
**Ribonucleoprotein complex** (1990904)	678	9	24.10	0.37	2.4 × 10^-2^
**Nucleolus** (0005730)	488	5	17.35	0.29	2.8 × 10^-2^
**Plant-type vacuole** (0000325)	787	8	27.97	0.29	6.7 × 10^-4^
**Chloroplast envelope** (0009941)	601	5	21.36	0.23	2.2 × 10^-3^
**Endosome** (0005768)	407	3	14.47	0.21	2.4 × 10^-2^
**Trans-Golgi network** (0005802)	281	1	9.99	0.1	3.4 × 10^-2^
**Cytosolic ribosome** (0022626)	294	1	10.45	0.1	1.9 × 10^-2^
**Thylakoid** (0009579)	533	1	18.50	0.05	2.4 × 10^-5^

## Data Availability

Data are contained within the article and Appendix A.

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
