# Peer review of "Scrutinizing the Impact of Alternating Electromagnetic Fields on Molecular Features of the Model Plant Arabidopsis thaliana"

_ijerph, 2022, doi:10.3390/ijerph19095144_

Round 1
Reviewer 1 Report
Please see the suggestions in the attached file.

Reviewer 2 Report
Overall, the research aims to determine the effect of EMF on Arabidopsis plants and looks for changes in the phenotype, transcriptome, metabolome and photosynthetic performance of 30 days old plants. In conclusion the authors claim to find no visible phenotype changes after 48h of EMF treatment but do claim to show changes in gene expression, metabolome and photosynthetic performance.
Overall, I found the research question of great importance and thought that the plan and methods used were well executed. The paper is well written and presented and the data and analysis appear scientifically sound.
The main criticism I have is addressed by the authors in the conclusion regarding the amount of light used in comparison to natural conditions and as they suggest it might be a good idea to use higher fluence rates of light and perhaps higher levels of EMF to look for a more obvious phenotype in the future. Also, as the authors stated using 30-day plants may not give a clear phenotype after only 2 days treatment and so longer treatments were used from 7 day old to 21 day old but again they state no obvious phenotype was found. What exactly were the authors looking at in regard to phenotype changes and did the authors try any treatments on germinating seeds or plants younger than 7 days?
Overall, I would recommend the paper for publication with some minor corrections and points addressed.
1- In nature what levels of EMF could a plant or human encounter and how does this compare to the levels used in the study?
2-Why was 48hrs of EMF treatment used, what is the rationale for this?
3-When looking for differences in phenotype what exactly did you look for i.e. fresh weight/dry weight, surface area etc. In Fig S1 it appears to me that your 21 day treated plants look a bit stressed and discoloured. What about the other repeats do they look the same? What exactly was the phenotypically analysis done on these plants and the other plants shown in FigS1.?
4- Fig 2, the top diagram is quite small and hard to read the hand writing
Round 2
Reviewer 1 Report
I had suggested RNASeq be repeated and maybe scrutinize a higher level of EMF to assess phenotype effect as some other studies show a significant effect. Please consider this if this research is carried out in future projects. the phenotypic set-up and RNA-Seq data as presented in this work are not up to the mark.
Author Response
Dear editor, dear Ms. Sun,
thanks for the editor's supportive decision. We are a little bit puzzled, because the referee did not make any comments for improvement, but suggestions for future work? Based on this advice, no changes must be made in the mansucript. Did we miss anything
Thanks a lot again and all the best
Karl-Josef Dietz